

# Dose-sparing effect of lapatinib co-administered with a high-fat enteral nutrition emulsion: preclinical pharmacokinetic study

Junfeng Zhu, Gaoqi Xu, Dihong Yang, Yu Song, Yinghui Tong, Sisi Kong, Haiying Ding and Luo Fang

Department of Pharmacy, Zhejiang Cancer Hospital, Hangzhou, China
Hangzhou Institute of Medicine (HIM), Chinese Academy of Sciences, Hangzhou, China

## ABSTRACT

**Background**. Lapatinib is an oral small-molecule tyrosine kinase inhibitor indicated for advanced or metastatic HER2-positive breast cancer. In order to reduce the treatment cost, a high-fat enteral nutrition emulsion TPF-T was selected as a dose-sparing agent for lapatinib-based therapies. This study aimed to investigate the effect of TPF-T on lapatinib pharmacokinetics.

**Methods**. First, a simple and rapid liquid chromatography tandem mass spectrometry (LC–MS/MS) method was developed to quantitatively evaluate lapatinib in rabbit plasma. The method was fully validated according to the China Pharmacopoeia 2020 guidance. Rabbits and rats were chosen as the animal models due to their low and high bile flows, respectively. The proposed LC–MS/MS method was applied to pharmacokinetic studies of lapatinib, with or without TPF-T, in rabbit and rat plasma.

**Results**. The LC–MS/MS method revealed high sensitivity and excellent efficiency. In the rabbit model, co-administration with TPF-T resulted in a 32.2% increase in lapatinib exposure. In the rat model, TPF-T had minimal influence on the lapatinib exposure. In both models, TPF-T was observed to significantly elevate lapatinib concentration in the absorption phase.

**Conclusion**. Co-administration with TPF-T had a moderate effect on increasing exposure to lapatinib. Dose sparing using a high-fat liquid diet is potentially feasible for lapatinib-based therapies.

Corresponding author
Luo Fang, fangluo@zjcc.org.cn

## INTRODUCTION

Lapatinib is an oral small-molecule tyrosine kinase inhibitor of the epidermal growth factor receptor (EGFR) and Erb-B2 receptor tyrosine kinase 2 (ERBB2 or HER2). It competitively blocks the ATP-binding of the receptors' intracellular tyrosine kinase domain, preventing the phosphorylation of receptors and inhibiting tumor cell proliferation (*Dong et al., 2022*; *Yu et al., 2022*). The combination of lapatinib with capecitabine has been approved for treating advanced or metastatic HER2-positive breast cancer patients who have progressed following standard therapy, including anthracyclines, taxanes, and

trastuzumab (*de Azambuja et al., 2014*). A clinical trial in women with HER2-positive metastatic breast cancer revealed that the combination therapy improved progression time without increased severe side effects (*Geyer et al., 2006*). However, the treatment cost of lapatinib is approximately €18,682 based on the average treatment duration (*Berghuis et al., 2018*). The high price of lapatinib is a potential barrier to its addition to the WHO Model List of Essential Medicines and is a heavy medical burden for cancer patients (*Hill et al., 2016*).

The dose-sparing strategy is a practical approach of combination with other low-toxicity agents to efficiently reduce the dose requirement of therapeutic drugs and consequently reduce healthcare costs for patients (*Cohen et al., 2014*; *Lee et al., 2020*). By accelerating absorption or inhibiting metabolism and excretion, the dose-sparing agent could significantly increase exposure to therapeutic drugs. For example, the Wuzhi capsule is a traditional Chinese medicine preparation commonly prescribed to alleviate the hepatotoxicity of tacrolimus in China (*Chen et al., 2021*; *Zhang et al., 2018*). It inhibits the metabolism of tacrolimus by inhibiting CYP3A4/5 (*He et al., 2022*; *Jackson et al., 2017*). Clinical pharmacokinetic studies revealed that the co-administration of Wuzhi capsule could significantly increase systemic exposure to tacrolimus, thus leading to an approximate dose reduction of 50% in patients using tacrolimus (*Cheng et al., 2021*; *Jing et al., 2021*; *Xin et al., 2011*). Therefore, searching for a dose-sparing agent to reduce the lapatinib dose is greatly beneficial for clinical cancer treatment.

It has been observed that administration with a high-fat diet significantly increases the exposure to lapatinib in cancer patients (*Devriese et al., 2014*; *Koch et al., 2009*). The high-fat diet was speculated to cause an increase in the quantity of bile salt released into the duodenum. Consequently, the solubility of lapatinib was enhanced by its incorporation into micelles formed by bile salts, which further increased its solubility and permeability into enterocytes, leading to a higher bioavailability (*Koch et al., 2009*). Based on the pharmacokinetic studies and predicted exposure, the co-administration of a high-fat diet might be a strategy to reduce the lapatinib dose (*Ratain & Cohen, 2007*). Nevertheless, the unpredictability of diet, such as the different eating habits among patients and variations of a patient on different days, could influence the consistent therapeutic exposure to lapatinib (*Rahman et al., 2007*).

High-fat enteral nutrition emulsion is widely used as a nutritional supplement in patients with tumors (*Li et al., 2011*; *Li et al., 2021*). As a standardized high-fat liquid diet, it is a promising dose-sparing agent in lapatinib-based therapies. The high-fat enteral nutrition emulsion offers many advantages over food, including fixed composition (caloric intake and fat, protein, and carbohydrate content), easy administration, controlled quality and quantity, as well as sufficient safety. This study was performed to investigate the effect of a high-fat enteral nutrition emulsion (TPF-T) on the pharmacokinetics of lapatinib in rabbits and rats using liquid chromatography tandem mass spectrometry (LC–MS/MS). To our knowledge, no work has been reported on the pharmacokinetic changes of lapatinib when administered simultaneously with a high-fat liquid diet.

## MATERIALS & METHODS

### Chemicals and reagents

The reference standards for lapatinib (≥98% purity) and dasatinib (≥98% purity) were purchased from Sigma-Aldrich (St. Louis, MO, USA). The lapatinib ditosylate tablets (250 mg per tablet) used for the animal experiments were produced by Glaxo Operations UK Limited (Ware, Herts, UK). TPF-T was produced by FRESENIUS KABI SSPC (Wuxi, China), with caloric distribution of 18% protein, 50% fat, and 32% carbohydrate. Detailed nutrient formulation of TPF-T can be found in the Supplementary materials (Table S1). Formic acid was purchased from Aladdin (Shanghai, China), whereas HPLC-grade methanol and acetonitrile were obtained from Merck (Darmstadt, Germany). Ultrapure water was prepared using a Milli-Q A10 water purification system (Millipore, Billerica, MA, USA).

### Standard and sample preparation

*Preparation of stock solutions and working solutions*

Two sets of lapatinib stock solutions were prepared by independently weighing the lapatinib and dissolving it in methanol to achieve a target concentration of 1.00 mg/mL. One was used for preparing calibration standards and the other for preparing the quality control (QC) samples. A stock solution of dasatinib (internal standard, IS) was prepared in methanol at 1.00 mg/mL. All the stock solutions were stored at −20 °C.

The working solutions of lapatinib for calibration standards were prepared by serial dilution of the stock solution using methanol-water (50:50, v/v) to achieve final concentrations of 20,000, 16,000, 10,000, 4,000, 2000, 500, 200, and 100 ng/mL. Lapatinib working solutions at 16,000, 10,000, 200, and 100 ng/mL for the QC samples and the IS working solution at 15,000 ng/mL were prepared similarly.

*Preparation of calibration standards and QC samples*

The calibration standards of lapatinib were prepared by spiking 5 μL of each working solution with 95 μL of blank plasma to achieve final concentrations of 1,000, 800, 500, 200, 100, 25, 10, and five ng/mL. QC samples at 800, 500, 10, and five ng/mL were prepared in the same manner as the calibrators.

*Sample preparation*

A simple protein precipitation method was used to extract lapatinib from the calibration standards, QC samples, and all the plasma samples. The IS working solution was diluted using acetonitrile to a final concentration of 150 ng/mL. Protein precipitation was initiated by adding 150 μL of acetonitrile solution containing the IS to 50 μL of each sample. After vortexing for 30 s, the mixed solution was centrifuged at 13,000 rpm for 10 min. The supernatant was transferred to an autosampler vial for LC–MS/MS analysis.

### Chromatographic and mass spectrometry conditions

LC–MS/MS analysis was carried out using a Waters XEVO TQ-S system, consisting of a Waters Acquity UPLC system, a triple quadrupole mass spectrometer, and an electrospray ionization (ESI) source as the interface. Chromatographic separations were performed on

an Agilent Eclipse XDB-C18 column (2.1 mm × 100 mm, 3.5 μm) using a gradient mobile phase consisting of 0.1% formic acid aqueous solution (A) and methanol (B) at a flow rate of 0.3 mL/min. The gradient elution program was as follows: 0–0.30 min, 10% B; 0.30–1.30 min, 10%–90% B; 1.30–2.70 min, 90% B; 2.70–2.71 min, 90%–10% B; 2.71–3.00 min, 10% B. The column was maintained at 40 °C, and the injection volume was 10 μL.

The experimental parameters of the mass spectrometer were: ionization mode, positive; capillary voltage, 3.2 kV; desolvation temperature, 600 °C; desolvation gas flow rate, 700 L/h; and cone gas flow rate, 100 L/h. Lapatinib and IS were determined in the multiple reaction monitoring (MRM) mode. The MRM transition for lapatinib quantification was $m/z$ 580.91 → 364.97, with corresponding cone voltage and collision energy of 96 V and 34 eV, respectively. The MRM transition for IS determination was $m/z$ 488.00 → 400.91, with corresponding cone voltage and collision energy of 88 V and 28 eV, respectively. Data were acquired and processed using the MassLynx software (Waters, version 4.1) provided with the instrument.

## Method validation

The method for determining lapatinib in rabbit plasma was validated according to the guidelines for quantitative analysis of biological samples (*Chinese Pharmacopoeia Commission, 2020*). Due to considerations of efficiency and cost, we did not specifically validate the quantification of lapatinib in rat plasma.

### Selectivity

Selectivity was evaluated by comparing the chromatograms from six different lots of blank rabbit plasma with spiked samples at lower limit of quantification (LLOQ). The acceptance criteria were: peak areas at the elution times of lapatinib and IS in the blank sample were no more than 20% of the lapatinib response and 5% of the IS response in the LLOQ sample.

### Linearity and LLOQ

Eight levels of calibration standards with lapatinib concentrations ranging from 5 to 1,000 ng/mL were prepared to evaluate the linearity. Calibration curves ($n = 3$) were established by plotting the peak area ratios of lapatinib to IS *versus* the nominal concentration of lapatinib using a $1/x^2$ weighted linear regression. The LLOQ was defined as the lowest concentration point on the calibration curve, at which the precision and accuracy should not exceed 20% ($n = 5$). The back-calculated concentration of the other calibration standards should be within ± 15% deviation of the nominal concentration.

### Injection carry-over

Carry-over was estimated by injecting an upper limit of quantification (ULOQ) sample prior to the injection of a blank sample. The peak areas of lapatinib and IS in the blank samples should not exceed 20% of the lapatinib response and 5% of the IS response in the LLOQ sample.

### Intra- and inter-day precision and accuracy

For the evaluation of intra- and inter-day accuracy and precision, five replicates at four QC levels were analyzed on separate days. The accuracy was expressed as the relative

error (RE%), determined as the percentage bias between the measured and nominal concentrations. Precision was represented by the relative standard deviation (RSD%). The RE% and RSD% should not exceed $\pm$ 15% and 15%, respectively, for QC samples at low, medium, and high concentrations. For QC samples at the LLOQ, RE% and RSD% should not exceed $\pm$ 20% and 20%, respectively.

### Extraction recovery

The extraction recovery was evaluated by comparing the IS-normalized lapatinib peak areas of the samples spiked before and after extraction. Three QC levels at low, medium, and high concentrations were evaluated in five replicates.

### Matrix effect

The matrix effect was represented by the IS-normalized matrix factor, which was calculated by comparing the normalized peak area ratio of lapatinib to IS spiked after extraction with that dissolved in the pure solution. The QC levels at low and high concentrations were evaluated in six replicates.

### Stability

The stability of lapatinib in rabbit plasma was evaluated under the following conditions: (1) ice bath for 3 h, (2) $-80$ °C refrigeration for 14 days, (3) three freeze-thaw cycles, (4) ice bath for 3 h after sample preparation, and (5) autosampler room (10 °C) for 4 h after sample preparation.

## Pharmacokinetic study

Two experimental models (rabbit and rat) were used to investigate the effects of TPF-T on lapatinib pharmacokinetics. The protocol was approved by the Animal Ethics Committee of Zhejiang Cancer Hospital (ethical approval number: 2022-11-001), in compliance with the national guidelines for the care and use of laboratory animals. Twelve female New Zealand white rabbits (weighing 3–4 kg) and eight female Sprague-Dawley rats (weighing 180–220 g) were purchased from Zhejiang Vital River Laboratory Animal Technology Co., Ltd. (Jiaxing, China). The animals were acclimated for at least 3 days prior to the experiments. All animals were individually housed in cages (one rabbit or four rats per cage), and provided with full-value nutritional granulated feed and distilled water. The housing environment was maintained at a temperature of 23 $\pm$ 3 °C, relative humidity between 40%–70%, and subjected to a light-dark cycle of 12 h. After completion of the experiments, rabbits were anesthetized with sodium pentobarbital and euthanized by exsanguination, while rats were euthanized by $CO_2$ inhalation.

For the rabbit model, the rabbits were fasted overnight and provided with water only before they were randomly divided into two groups ($n = 6$). In group A, lapatinib (60 mg/kg) was administered by intragastric gavage with water. In group B, lapatinib (60 mg/kg) was administered $via$ intragastric gavage with TPF-T. Blood samples were collected from the marginal ear vein into EDTA anticoagulation tubes prior to dosing and at 0.25, 0.5, 1, 1.5, 2, 3, 4, 6, 8, 10, 12, 24, 48, and 72 h after dosing. After centrifugation at 3,000 rpm for 10 min, the supernatants of the blood samples were collected and stored at $-$ 80 °C for

further analysis. For the rat model, eight Sprague-Dawley rats were randomly divided into two groups ($n = 4$), and lapatinib was administered simultaneously with water or TPF-T. Blood samples were collected from tail snips and prepared in a manner consistent with the rabbit model.

### Data analysis

Pharmacokinetic parameters were calculated by non-compartmental methods using Drug and Statistics (DAS) 2.0. Statistical comparisons between two groups were performed using unpaired $t$-tests with two-tailed distributions using GraphPad Prism 8 (GraphPad, San Diego, CA, USA). Welch's correction was applied when there were significant differences in variances.

## RESULTS

### Method validation

Figure 1 presents the MRM chromatograms of blank rabbit plasma, blank rabbit plasma spiked with lapatinib and IS at the LLOQ concentration, and rabbit plasma collected after oral administration of lapatinib. Under the proposed conditions, lapatinib and IS were eluted at 2.09 and 1.94 min, respectively. No significant interference was observed at the elution times of lapatinib and IS in the blank sample. At the LLOQ level, a signal-to-noise ratio (S/N) of >5 was obtained for lapatinib. Lapatinib demonstrated good linearity in the range of 5–1,000 ng/mL with $r \geq 0.997$. The typical linear equation for the calibration curve was $y = 1.02805x + 1.26591$ (Fig. S1). The LLOQ of this method was five ng/mL with accuracy (RE) and precision (RSD) of <20% (Table S2). The peak area of lapatinib in the blank sample injected after the ULOQ sample exceeded 20% of the LLOQ responses. To mitigate the interference of the carry-over, additional blank sample injections were included after the samples with high concentrations. The mean accuracies of the intra-day QC samples ranged from −3.49 to 5.46%, with precisions ranging from 0.87 to 7.90%. The mean accuracies of the inter-day QC samples ranged from 0.92 to 7.56%, with precisions ranging from 2.99 to 7.98% (Table S2). The mean IS-normalized extraction recovery of lapatinib was in the range of 99.46–103.40% with RSD $\leq 2.90\%$, and the IS-normalized matrix factors of lapatinib at low and high concentration levels were $88.32 \pm 3.01$ and $88.70 \pm 1.03\%$, respectively (Table S3). The stability of lapatinib in rabbit plasma under specific conditions ranged from 97.20 to 124.03% (Table S4). The above results demonstrated that this method was specific, sensitive, and robust enough for the pharmacokinetic study of lapatinib.

### The effect of TPF-T on lapatinib pharmacokinetics in rabbit plasma

The plasma concentration–time curves of lapatinib after a single oral dose in rabbits with and without TPF-T are presented in Fig. 2A, and the pharmacokinetic parameters of lapatinib are summarized in Table 1. When co-administered with TPF-T, the $C_{max}$ of lapatinib decreased by 25.5% ($P = 0.27$), while the $AUC_{0-t}$ increased by 32.2% ($P = 0.17$). Although the differences in the pharmacokinetic parameters between both groups were not significant, the pharmacokinetic behavior of lapatinib changed dramatically when
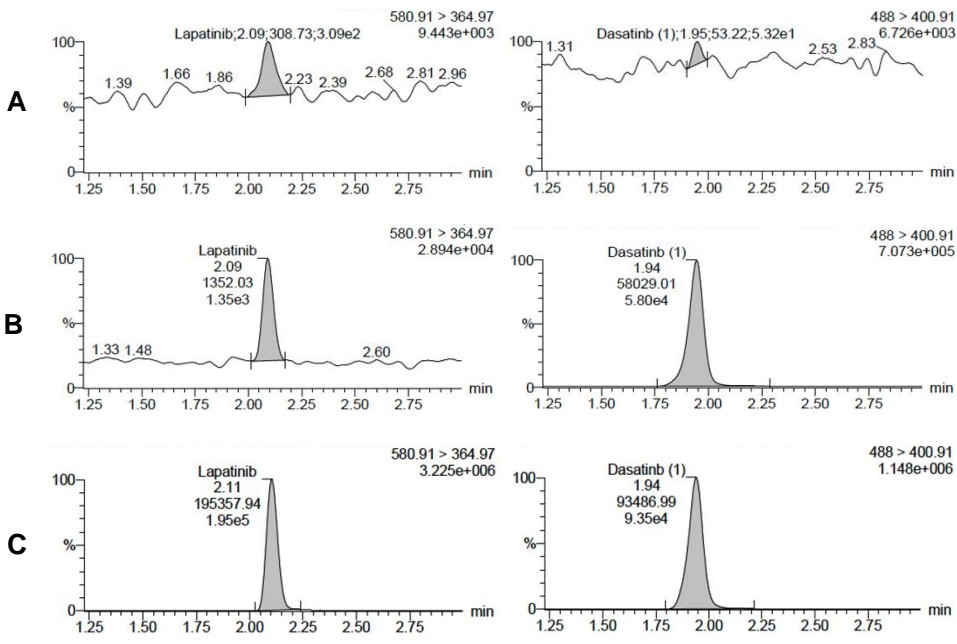

**Figure 1 Chromatograms of lapatinib (left) and IS (right) in different samples.** (A) Blank rabbit plasma, (B) blank rabbit plasma spiked with lapatinib and IS at LLOQ concentration, (C) rabbit plasma collected at 6 h after oral administration of lapatinib.

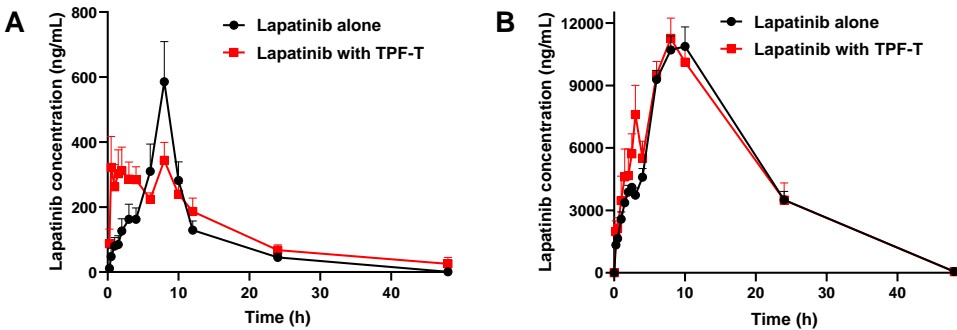

**Figure 2 Mean plasma concentration–time curves of lapatinib after a single oral dose of lapatinib with and without TPF-T.** (A) Rabbits, (B) rats. Data are expressed as mean ± SEM.

co-administered with TPF-T. For example, the plasma concentration of lapatinib at 0.5–4 h significantly increased when it was co-administered with TPF-T ($P < 0.05$). These results suggested that co-administration with TPF-T could partially increase the absorption of lapatinib in rabbits.

## The effect of TPF-T on lapatinib pharmacokinetics in rat plasma

The plasma concentration–time curves of lapatinib after a single oral dose in rats with and without TPF-T are presented in Fig. 2B, and the pharmacokinetic parameters of lapatinib

**Table 1 Pharmacokinetic parameters of lapatinib after a single oral dose of lapatinib (60 mg/kg) to rabbits with and without TPF-T.**

| Parameters | Lapatinib alone | Lapatinib with TPF-T |
|---|---|---|
| $C_{max}$ (ng/mL) | $606.1 \pm 280.3$ | $451.6 \pm 157.6$ |
| $T_{max}$ (h) | $7.7 \pm 1.5$ | $4.6 \pm 4.5$ |
| $t_{1/2z}$ (h) | $3.9 \pm 2.9$ | $7.4 \pm 4.6$ |
| $MRT_{0-t}$ (h) | $9.2 \pm 1.1$ | $11.9 \pm 4.1$ |
| $MRT_{0-\infty}$ (h) | $10.5 \pm 1.7$ | $16.3 \pm 7.5$ |
| $V_z/F$ (L/kg) | $76.8 \pm 51.3$ | $113.3 \pm 65.1$ |
| $CL_z/F$ (L/h/kg) | $15.3 \pm 5.2$ | $11.2 \pm 3.7$ |
| $AUC_{0-t}$ (ng/mL*h) | $4,206.3 \pm 1,531.9$ | $5,562.0 \pm 1,657.5$ |
| $AUC_{0-\infty}$ (ng/mL*h) | $4,359.3 \pm 1,534.5$ | $5,791.4 \pm 1,730.5$ |

**Notes.**
Data are expressed as mean $\pm$ SD ($n = 6$).
$C_{max}$, maximum concentration; $T_{max}$, time to reach $C_{max}$; $t_{1/2z}$, terminal elimination half-life; MRT, mean residence time; $V_z/F$, volume of distribution; $CL_z/F$, clearance; AUC, area under the curve.

**Table 2 Pharmacokinetic parameters of lapatinib after a single oral dose of lapatinib (100 mg/kg) to rats with and without TPF-T.**

| Parameters | Lapatinib alone | Lapatinib with TPF-T |
|---|---|---|
| $C_{max}$ (ng/mL) | $11,052.3 \pm 1,729.0$ | $1,1259.8 \pm 1,947.2$ |
| $T_{max}$ (h) | $8.5 \pm 1.0$ | $8.0 \pm 0.0$ |
| $t_{1/2z}$ (h) | $6.2 \pm 0.1$ | $6.2 \pm 0.3$ |
| $MRT_{0-t}$ (h) | $13.5 \pm 0.7$ | $13.1 \pm 1.0$ |
| $MRT_{0-\infty}$ (h) | $13.5 \pm 0.7$ | $13.2 \pm 1.0$ |
| $V_z/F$ (L/kg) | $4.2 \pm 0.7$ | $4.4 \pm 0.8$ |
| $CL_z/F$ (L/h/kg) | $0.5 \pm 0.1$ | $0.5 \pm 0.1$ |
| $AUC_{0-t}$ (ng/mL*h) | $212,984.6 \pm 33,405.3$ | $212,986.3 \pm 517,15.5$ |
| $AUC_{0-\infty}$ (ng/mL*h) | $213,064.7 \pm 33,421.8$ | $213,079.9 \pm 51,757.6$ |

**Notes.**
Data are expressed as mean $\pm$ SD ($n = 4$).

are summarized in Table 2. Except for the elevated lapatinib concentration at 2.5–3 h, TPF-T had minimal influence on the lapatinib pharmacokinetics in rats.

## DISCUSSION

Compared to other oral tyrosine kinase inhibitors, food has more significant effect on lapatinib exposure in humans. Co-administration with food, especially a high-fat diet, resulted in a 3.25-fold increase in the bioavailability of lapatinib (*Koch et al., 2009*). Therefore, we speculated that high level of fat plays a positive role in improving lapatinib bioavailability. Due to the unpredictability of daily diet, here a standardized high-fat liquid diet TPF-T was employed as a dose-sparing agent for lapatinib, and the effect of TPF-T on the pharmacokinetics of lapatinib was investigated.

Species with higher bile flow (for example, rats and dogs) were reported to experience the opposite effect of food on lapatinib's bioavailability (*Koch et al., 2009*). The solubilization capacity of these species is greater, favoring fecal excretion over absorption and resulting in a higher recovery of unmetabolized lapatinib in feces. So rabbits were selected as the primary experimental model due to their low bile flow, similar to that of humans. In spite of this, a rat model was also used to investigate the effect of TPF-T on lapatinib pharmacokinetics. The plasma concentration of lapatinib was evaluated using LC–MS/MS method. LC–MS/MS is the most commonly used method in pharmacokinetic studies due to its excellent sensitivity and efficiency (*Wang et al., 2021*; *Yang et al., 2021*), while several methodologies using LC–MS/MS to quantify lapatinib in human or rat plasma have been published (*Alrobaian et al., 2022*; *Karbownik et al., 2020*; *Karbownik et al., 2018*; *Li, Zhao & Zhao, 2022*), there's no literature describing the quantification of lapatinib in rabbits. Here a sensitive and robust LC–MS/MS method was developed to quantify lapatinib in rabbit plasma. This method, in principle, could also quantify lapatinib in rat plasma. However, no specific validation has been undertaken due to efficiency and economy.

In the rabbit model, co-administration with TPF-T resulted in a 32.2% increase in lapatinib exposure ($P = 0.17$). Meanwhile, TPF-T significantly elevated lapatinib plasma concentration during the absorption phase. It is speculated that the high fat content in TPF-T stimulated bile secretion and consequently enhanced lapatinib absorption. However, compared to the influence of high-fat food on the bioavailability of lapatinib in humans (325% increase), the effect of TPF-T on increasing exposure to lapatinib in rabbits was moderate. This was likely attributed to the limited efficacy of a liquid diet in delaying gastric emptying and reducing small intestinal motility. Meanwhile, food may alter lapatinib bioavailability by changing the gastrointestinal pH, increasing splanchnic blood flow, affecting metabolism, and many other factors (*Huh et al., 2023*; *Piscitelli et al., 2023*; *Vertzoni et al., 2019*). In the rat model, TPF-T had minimal influence on the lapatinib pharmacokinetics probably because rats had higher bile flow. When the concentration of bile salts exceeds 15 mmol/L, the diffusion of micelles through the unstirred water layer to the enterocyte membrane is reduced, consequently limiting the effect of bile acid solubilization on lapatinib bioavailability (*Bakatselou, Oppenheim & Dressman, 1991*). Nevertheless, TPF-T could partially increase the plasma concentration of lapatinib in the absorption phase in both rabbit and rat models. From another perspective, the influence of TPF-T on the pharmacodynamics of lapatinib cannot be simply predicted from its bioavailability. This is because the maintenance time of lapatinib concentration in therapeutic window may be lengthened during combination therapy. A dose-sparing approach using a high-fat oral liquid may be feasible for lapatinib-based therapies. Additionally, the concept of dose sparing through the utilization of a high-fat liquid diet is innovative in the field of drug development and holds promising potential for clinical applications.

The study had several limitations. First, it may fail to accurately reflect the effect of TPF-T on lapatinib pharmacokinetics in clinic because of differences in physiology between experimental animals and humans. Meanwhile, food effects mainly result from a combination of factors. *In vitro* dissolution tests should be further performed to study

the physical/chemical interaction between TPF-T and lapatinib. The alternations of TPF-T on gastric emptying, gastrointestinal pH, bile flow, and splanchnic blood flow are also needed. In addition, the influence of TPF-T on the pharmacodynamics of lapatinib should be investigated further.

## CONCLUSIONS

In this study, a standardized high-fat liquid diet TPF-T was employed as a dose-sparing agent for lapatinib. We successfully developed a simple and rapid LC–MS/MS method for the quantitative evaluation of lapatinib in rabbit plasma. The method was validated according to the China Pharmacopoeia 2020 guidance, demonstrating high sensitivity and excellent efficiency. The proposed method was applied for the first time to study the effect of a high-fat liquid diet on lapatinib pharmacokinetics in rabbit and rat plasma. Co-administration with TPF-T resulted in an approximately 30% increase in the exposure to lapatinib in rabbits. Additionally, TPF-T could significantly elevate lapatinib concentration in the absorption phase. Therefore, a dose sparing approach using a high-fat liquid diet is potentially feasible for lapatinib-based therapies.

## ACKNOWLEDGEMENTS

The authors thank Prof. Qinjie Weng and Prof. Lushan Yu at Zhejiang University for their help with the animal experiments and method development.

### Funding
The authors received no funding for this work.

### Competing Interests
The authors declare there are no competing interests.

### Author Contributions
- Junfeng Zhu conceived and designed the experiments, performed the experiments, analyzed the data, prepared figures and/or tables, authored or reviewed drafts of the article, and approved the final draft.
- Gaoqi Xu performed the experiments, analyzed the data, prepared figures and/or tables, authored or reviewed drafts of the article, and approved the final draft.
- Dihong Yang performed the experiments, analyzed the data, prepared figures and/or tables, authored or reviewed drafts of the article, and approved the final draft.
- Yu Song performed the experiments, authored or reviewed drafts of the article, and approved the final draft.
- Yinghui Tong conceived and designed the experiments, performed the experiments, authored or reviewed drafts of the article, and approved the final draft.
- Sisi Kong performed the experiments, authored or reviewed drafts of the article, and approved the final draft.

- Haiying Ding performed the experiments, authored or reviewed drafts of the article, and approved the final draft.
- Luo Fang conceived and designed the experiments, analyzed the data, authored or reviewed drafts of the article, and approved the final draft.

## Animal Ethics

The following information was supplied relating to ethical approvals (i.e., approving body and any reference numbers):

The protocol was approved by the Animal Ethics Committee of Zhejiang Cancer Hospital.

## Data Availability

The raw data are available in the Supplementary Files.

## Supplemental Information

Supplemental information for this article can be found online at http://dx.doi.org/10.7717/peerj.16207#supplemental-information.

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
