# Peer review of "Dose-sparing effect of lapatinib co-administered with a high-fat enteral nutrition emulsion: preclinical pharmacokinetic study"

_PeerJ, doi:10.7717/peerj.16207_

## Round 0.1 · original submission · Minor Revisions

This study raises the possibility of reducing the costs of breast cancer treatment with lapatinib, which is quite important issue. Please answer carefully all the questions of the reviewers with special attention on those related to the validity of the findings.

Reviewer 1 ·

Basic reporting

As per the Title which explains the research intent - The article is fully aligned with it.
The research work focuses on breast cancer oral small-molecule tyrosine kinase inhibitor Lapatinib dose, How to make the treatment cost-effective.

The team has done research on high-fat enteral nutrition emulsion TPF-T as a dose-sparing agent for lapatinib-based therapies to study its influence on lapatinib exposure.
English used in the manuscript was very professional.
Literature references were of the required field.

The figures, and raw data which were shared were as per the experiments performed and relevant.

Experimental design

The experimental design was very well explained and all the activities were given in detail.
The sample preparation, Internal standard used, The instrument technical specification, and mobile phase column used were very well described.
The method validation parameters were as per the requirement of the experiment.
The stability of Lapitinib in serum was also studied.

In the Pharmacokinetic study, Two experimental models (rabbit and rat) were used to investigate the effects of TPF-T on lapatinib in rabbits and rats.

When co-administered with TPF-T, the Cmax of lapatinib decreased by 25.5% (P = 0.27), but the AUC0-t increased by 32.2% the plasma concentration of lapatinib had increased to 0.5–4 h significantly.

It was well explained the fat diet TPF-T showed significant absorption of Laptininb for a longer duration which would certainly reduce the cost of treatment, but still further investigation on Humans is required.

Validity of the findings

The research work focuses on reducing the cost of treatment for breast cancer, but initially, the experiment was designed for rabbit and rat models.
The results were well validated as per China Pharmacopoeia 2020.

The experimental data provided were consistent. All the required parameters for pharmacokinetics were considered. The conclusion of the experiment was very well stated.

Additional comments

Only one correction,
the reference's author names should not be in bold letters.

Reviewer 2 ·

Basic reporting

This manuscript reports a study that explores an interesting topic, with significant potential to impact clinical practice. The topic is of great importance, given the potential for a dose-sparing strategy to enhance the cost-effectiveness and accessibility of lapatinib treatment. Nonetheless, there are some minor areas where the manuscript could be improved to strengthen the presentation and interpretation of the study findings.
Major Points:
• The manuscript provides an in-depth background on lapatinib and the value of a dose-sparing strategy. However, the link between these two concepts could be further clarified. The study's objectives should be clearly and concisely stated in relation to the detailed background.
• The introduction could benefit from a more explicit connection between the problem identified (high cost of lapatinib) and the proposed solution (dose-sparing strategy with a high-fat enteral nutrition emulsion). Additionally, the manuscript would be greatly strengthened by a more thorough and broader discussion (cross-modality, cross-therapeutic area) on the potential of high-fat diets in pharmacokinetics and dose-sparing strategies.
Minor Points:
• The manuscript is generally well-written but a few areas could be improved for better grammar and readability. Below are some suggested edits:
o Line 26 to 27: "Rabbit and rat were chosen as the animal models with low and high bile flow, respectively." This sentence may be better phrased. Consider: "Rabbits and rats were chosen as the animal models due to their low and high bile flows, respectively."
o Line 98 to 99: The sentence "Two sets of lapatinib stock solutions were prepared in methanol using two independent weightings at a target concentration of 1.00 mg/mL." may be rephrased as "Two sets of lapatinib stock solutions were prepared by independently weighing the lapatinib and dissolving it in methanol to achieve a target concentration of 1.00 mg/mL."
o Line 140 to 142: The sentence "Considering its efficiency and economy, specific validation of lapatinib quantification in rat plasma has not been undertaken," could be reworded for better clarity: "Due to considerations of efficiency and cost, we did not specifically validate the quantification of lapatinib in rat plasma."
o Line 179 to 199: The section "Pharmacokinetic study" contains some long sentences that could be broken up for improved readability. For instance, the sentence "Twelve female New Zealand white rabbits (weighing 3.4 kg) and eight female Sprague-Dawley rats (weighing 180.220 g) were purchased from Zhejiang Vital River Laboratory Animal Technology Co., Ltd. (Jiaxing, China) and acclimated for at least 3 days prior to the experiments." could be broken into two: "Twelve female New Zealand white rabbits (weighing 3.4 kg) and eight female Sprague-Dawley rats (weighing 180.220 g) were purchased from Zhejiang Vital River Laboratory Animal Technology Co., Ltd. (Jiaxing, China). The animals were acclimated for at least 3 days prior to the experiments."
o Line 199 to 200: In the sentence "Blood samples were collected from tail snips and prepared similarly to the rabbit model," it would be helpful to specify that this statement is on the rat model to improve clarity.
o Line 203: In the sentence "Pharmacokinetic parameters were calculated by non-compartmental methods using DAS 2.0," it would be helpful to provide full names at least once for all abbreviations for the benefit of the readers, e.g., "Drug and Statistics (DAS) 2.0."
o Line 236: Use "rabbits" instead of "rabbit" for better grammar.
o Line 258 to 261: The sentence "Several LC-MS/MS methodologies have been published to quantify lapatinib in human or rat plasma (Alrobaian et al., 2022; Karbownik et al., 2020; Karbownik et al., 2018; Li et al., 2022), but no literature has described the quantification of lapatinib in rabbits." could be restructured for better readability: "While several methodologies using LC-MS/MS to quantify lapatinib in human or rat plasma have been published (Alrobaian et al., 2022; Karbownik et al., 2020; Karbownik et al., 2018; Li et al., 2022), there's no literature describing the quantification of lapatinib in rabbits."
o Line 262 to 263: In the sentence "In principle, lapatinib in rat plasma was also quantifiable by this method," to improve readability, consider: "This method, in principle, could also quantify lapatinib in rat plasma."
o Line 279 to 281: In the sentence "From another point of view, the influence of TPF-T on the pharmacodynamics of lapatinib could not be simply predicted from its bioavailability because the maintenance time of lapatinib concentration in therapeutic window may be lengthened during combination therapy," consider breaking this into two sentences to improve readability: "From another perspective, the influence of TPF-T on the pharmacodynamics of lapatinib cannot be simply predicted from its bioavailability. This is because the maintenance time of lapatinib concentration in the therapeutic window may be lengthened during combination therapy."
o Line 282: In the sentence "Dose sparing using a high-fat oral liquid is potentially feasible for lapatinib-based therapies," the meaning could be clearer if rephrased: "A dose-sparing approach using a high-fat oral liquid may be feasible for lapatinib-based therapies." Similarly, Line 299 to 300: "Therefore, dose sparing using a high-fat liquid diet is potentially feasible for lapatinib-based therapies." may be rephrase as "Therefore, a dose sparing approach using a high-fat liquid diet is potentially feasible for lapatinib-based therapies."

Experimental design

Major Points:
• N/A
Minor Points:
• Although the use of rabbits and rats as animal models is mentioned, the rationale for their selection is not clear (though briefly mentioned in discussion section, to represent low and high bile flow models, Line 252 and 254). Providing more explanations on the choice of animal models would strengthen the design and improve reader understanding.
• A more detailed description of the high-fat enteral nutrition emulsion used would help readers understand why it was chosen and how it could potentially influence lapatinib pharmacokinetics.

• In the method validation section (Line 139 to 158), while the authors have done an excellent job detailing the process, I would suggest adding a figure that shows the calibration curve of lapatinib. This would allow for visual representation and could enhance the reader's understanding of the process. Additionally, providing explicit calculation or information on the Lower Limit of Quantification (LLOQ) and Upper Limit of Quantification (ULOQ) would be advantageous for the transparency of the methodology. These points, though minor, would serve to enhance the reproducibility and depth of this valuable study.

Validity of the findings

Major Point:
• I recommend a cautious interpretation of the findings, particularly where non-significant differences in pharmacokinetic parameters are discussed. Additional statistical analysis should be performed to evaluate biological variation, analytical method variation, and other potential sources of experimental errors, to confirm these findings.
Minor Points:
• While potential mechanisms (e.g., difference in TPF-T influence on lapatinib pharmacokinetics due to bile flow difference) for the observed effects are proposed, it's not clear whether these are hypotheses or were tested in this study. Clarity on this point would improve the discussion.
• It is encouraging to see the limitations of the study acknowledged. However, it would be beneficial to discuss potential strategies to address these limitations in future research.

Additional comments

The study presents a potentially significant contribution to the field of cancer treatment, specifically for patients undergoing lapatinib treatment. However, it would benefit from further elaboration on the connections between the problem (high cost of lapatinib) and the proposed solution (dose-sparing with a high-fat enteral nutrition emulsion).
Further, a clearer rationale for the experimental design, comprehensive statistical analysis, and a detailed explanation of the results are recommended to bolster the study's validity. A minor revision considering these points would add considerably to the manuscript.

·

Basic reporting

Mo comment

Experimental design

Method was validated using rabbit plasma and applied to rat plasma also. However no partial validation was performed for change in species. Hence there is no validation support available for use of method on rat plasma.
Is there any criteria used for selection of1x2 weighted linear regression.

Validity of the findings

In Table 5; Stability (%) of lapatinib in rabbit plasma under specific condition (ice bath for 3 h) at 10 ng/mL is more than 124 %. At LQC level it is expected to be within 85 to 115%. No explanation for this variation is provided.
Three precision and accuracy batches were analysed but intra-day precision and accuracy were presented from batch 3 only, which was having best result.
Clarification on area observed in blank rabbit plasma and blank plasma spiked with LLOQ concentration is required (Fig 1).

Additional comments

I did not find any novelty in the work. However, experiments were performed as per internationally followed procedures. Animal ethics committee approval was taken.
Influence of high-fat food on the bioavailability of lapatinib in humans (325% increase) is already reported. However, the effect of TPF-T on increasing exposure to lapatinib in rabbits was found to be moderate only in this study.

---

## Round 0.2 · accepted · Accept

All the reviewer´s comments were adequately addressed and the manuscript is ready for publication.

Reviewer 2 ·

Basic reporting

No more comments.

Experimental design

No more comments.

Validity of the findings

No more comments.

Additional comments

The authors have comprehensively addressed my previous comments, and the quality of their manuscript revision is commendable. I now endorse the publication of the manuscript.

·

Basic reporting

No further comment

Experimental design

No further comment

Validity of the findings

No further comment